# Pentaglobin^®^ Efficacy in Reducing the Incidence of Sepsis and Transplant-Related Mortality in Pediatric Patients Undergoing Hematopoietic Stem Cell Transplantation: A Retrospective Study

**DOI:** 10.3390/jcm9051592

**Published:** 2020-05-24

**Authors:** Carlone Giorgia, Torelli Lucio, Maestro Alessandra, Zanon Davide, Barbi Egidio, Maximova Natalia

**Affiliations:** 1Department of Medicine, Surgery and Health Sciences, University of Trieste, Piazzale Europa 1, 34127 Trieste, Italy; giorgiacarlone@gmail.com (C.G.); torelli@units.it (T.L.); egidio.barbi@burlo.trieste.it (B.E.); 2Institute for Maternal and Child Health - IRCC Burlo Garofolo, via dell’Istria 65/1, 34137 Trieste, Italy; alessandra.maestro@burlo.trieste.it (M.A.); davide.zanon@burlo.trieste.it (Z.D.)

**Keywords:** hematopoietic stem cell transplantation, pediatric patients, aplasia, Pentaglobin^®^, infection-related mortality rate

## Abstract

The 12-month mortality rate in patients undergoing hematopoietic stem cell transplantation (HSCT) remains high, especially with respect to transplant-related mortality (TRM), which includes mortality due to infection complications through the aplasia phase. The aim of this study was to determine whether the administration of Pentaglobin^®^ could decrease TRM by lowering sepsis onset or weakening sepsis through the aplasia phase. One hundred and ninety-nine pediatric patients who had undergone HSCT were enrolled in our retrospective study. The patients were divided into two groups: the Pentaglobin group, which had received Pentaglobin^®^ in addition to the standard antibiotic treatment protocol established for the aplasia phase, and the Control group, which received only the standard treatment. As compared to the control group outcome, Pentaglobin^®^ led to a significant decrease in the days of temperature increase (*p* < 0.001) and a reduced infection-related mortality rate (*p* = 0.04). In addition, the number of antibiotics used to control infections, and the number of antibiotic therapy changes needed following first-line drug failure, were significantly lowered in the Pentaglobin group as compared to the control group (*p* < 0.0001). With respect to the onset of new infections following the primary infection detected, the Pentaglobin group showed a significant reduction for bacterial events, as compared to the control group (*p* < 0.03). Pentaglobin^®^ use in patients undergoing HSCT seems to produce a significant decrease in infection-associated TRM rate.

## 1. Introduction

Hematopoietic stem cell transplantation (HSCT) is the most efficient consolidation therapy in some hematologic malignancies such as acute lymphoblastic leukemia and acute myeloid leukemia. HSCT is also a potential therapy for patients with solid tumors, genetic, hematological and metabolic disorders, and primary immunodeficiency diseases [1,2,3,4]. Nevertheless, HSCT results in a variety of severe complications responsible for a high rate of morbidity and mortality in transplant recipients [5].

In recent years, progress has been achieved in transplantation medicine with respect to high-resolution donor-recipient human leukocyte antigen (HLA) matching, conditioning regimens for HSCT, graft-versus-host disease (GVHD), and strict infection control. This has improved clinical outcomes by decreasing the rate of transplant-related mortality (TRM) [6,7]. However, all scientific societies consider the current mortality rate, of about 10% of transplant recipients on average during a 12-month follow-up period, to still be too high, so they identified a new primary endpoint at a lower rate, which remained below 10% [8].

The main causes of HSCT failure consist of disease relapse, GVHD, and sepsis onset developed through bone marrow aplasia [9,10,11,12]. Notably, GVHD and sepsis are two different complications responsible for TRM following HSCT that share a common pathogenic mechanism, independent of the primary disease. The common pathogenic element consists of Gram-negative intestinal bacteria that produce an endotoxin, the lipopolysaccharide component of the Gram-negative outer bacterial membrane. This endotoxin is responsible for the main adverse side effects of bowel flora, specifically, loss of gut wall integrity and endotoxin translocation into the blood, which leads to cytokine deregulation causing fever, hemodynamic instability, and ultimately multiorgan failure (MOF). Endotoxins, releasing tumor necrosis factor (TNF), might induce septic shock and GVHD onset conditioning morbidity and TRM [10]. 

Any therapy that can decrease endotoxemia could, therefore, provide better control of morbidity in transplant recipients. One such therapeutic option might be Pentaglobin^®^ (Biotest Pharma GmbH, Dreieich, Germany), an IgM-enriched immunoglobulin preparation, whose IgM component is responsible for endotoxin antibody activity. Indeed, IgM is the main component of the primary antibody response, and its pentameric structure is responsible for superior efficacy in toxin neutralization and bacterial agglutination as compared with IgG antibodies [13]. Thus, IgM is thought to be an important component in the intravenous immunoglobulin preparation.

Our study has sought to determine whether the use of Pentaglobin^®^ as early adjuvant treatment for febrile and subfebrile pediatric patients undergoing HSCT improves clinical outcomes, decreasing the early TRM rate and beyond. Moreover, we conducted the current study to gather preliminary data in order to start a prospective multicenter randomized controlled trial, supervised by GITMO, to define the real efficacy of Pentaglobin^®^ in patients undergoing HSCT.

## 2. Materials and Methods

### 2.1. Study Design and Population

A retrospective single-center study was carried out at the Pediatric Transplant Center of the Institute for Maternal and Child Health—IRCCS “Burlo Garofolo,” Trieste, Italy. The Institutional Review Board of the IRCCS Burlo Garofolo (reference no. RC 27/18) and the Unique Regional Ethics Committee (reference no. 2620) approved the study protocol. All parents of the patients gave written consent for the collection and use of personal data for research purposes. 

The medical records of all patients who underwent allogeneic or autologous HSCT at our Center between January 2000 and June 2018 were analyzed. Inclusion criteria were: age of recipient <18 years at the time of transplantation, patients for whom this was their first transplantation, myeloablative conditioning regimen, and six-month follow-up minimum. Patients with documented bacterial or fungal infection at the time of HSCT, patients in antibiotic treatment for any reason at the time of HSCT, or patients with previous adverse reactions due to intravenous immunoglobulin infusion were excluded from the study. The flow chart shows the number of HSCT recipients who were screened, enrolled, and included in the final analyses (Figure 1).

Data were analyzed with respect to a variety of demographic and clinical variables, including sex, age in years, underlying disease, pre-transplant immunity status (namely number of lymphocytes and level of serum immunoglobulins), conditioning regimen, and donor type. In addition, the following information was collected for all patients: type of infection, number of days with temperature rise ≥37.3 °C (99.1 °F), number of fever days, duration of Pentaglobin^®^ administration, duration of antibiotic treatment, number of antibiotics used, number of antibiotics changed because of treatment failure and acute GVHD onset (any grade). 

The primary outcome evaluated was the possible difference in TRM in both groups after a six-month follow-up. Secondary outcomes included the comparison between the two groups in the number of fever days, number of infectious events, duration, and number of antibiotics used to control every single event. 

### 2.2. HSCT Procedure

All patients who underwent allogeneic HSCT were treated according to standard myeloablative protocols. In patients over two years of age with acute lymphoblastic leukemia (LLA), the myeloablative conditioning regimen preceding allogeneic HSCT was based on total-body irradiation (TBI), while in the remaining cases, a busulfan-based conditioning regimen was used. In both cases, conditioning also included high-dose cyclophosphamide (1800 mg/m^2^ for two consecutive days). In the case of matched unrelated donors, haploidentical or sibling donors, and patients with hemoglobinopathy, rabbit anti-thymocyte globulin (ATG) was used. GVHD prophylaxis was performed with calcineurin-inhibitor alone or associated with mycophenolate mofetil and prednisone, as previously described [13]. All patients who underwent autologous HSCT received myeloablative conditioning regimen according to the current protocols based on the underlying disease. Disease risk was defined according to diagnosis and disease stage [14]. Supportive care for GVHD and infectious disease prophylaxis, mucositis, and veno-occlusive disease (VOD) did not substantially change during the time reported in this study.

### 2.3. Pentaglobin^®^ Administration

Patients included in the study were divided into two groups. Those who had received Pentaglobin^®^, in addition to the standard antibiotic treatment protocol established for the aplasia phase, were identified by accessing pharmacy records. Only those whose first cycle of Pentaglobin^®^ administration started within 12 h from first body temperature rise ≥37.3 Celsius degrees (°C, 99.1 °F) were included in the Pentaglobin group. Patients who received started Pentaglobin^®^ administration more than 12 h after their first temperature rise were excluded from the study. Pentaglobin^®^ was administered at a dose of 5 mL/kg/day in continuous infusion for three days in most cases. Pentaglobin^®^ administration was shortened in case of neutrophil engraftment with complete resolution of infection symptoms or extended if symptoms persisted in patients with severe mucositis or documented infection. The second group (the Control group) underwent only the standard antibiotic treatment protocol for the aplasia phase which consisted in the use of a third-generation cephalosporin plus aminoglycoside, as first-line therapy; despite first-line empirical antibiotic therapy, patients who remained febrile after 48 h started vancomycin; whereas, if blood culture results were available, a specific antibiotic treatment was undertaken.

### 2.4. Statistical Analysis

Quantitative variables were reported as median value and range or using the median and interquartile range, whereas categorical variables were expressed as absolute value and percentage. Demographic and clinical characteristics of patients were compared using the chi-square test or Fisher’s exact test for categorical variables, whereas the Mann–Whitney rank-sum test or the Student’s *t*-test were used for continuous variables; in addition, we performed proportion tests to compare categorical and continuous variables, as appropriate. The primary endpoint, overall survival, and event-free survival were calculated according to the Kaplan–Meier method. Comparisons between different overall survival and event-free survival probabilities were performed using the log-rank test, whereas multivariate analysis was performed using logistic regression in order to adjust the association of overall survival with clinical and demographic variables. *p* < 0.05 was considered to be statistically significant, and statistical analysis was performed using R statistical software (R version 3.5.2, 2018 The R Foundation for Statistical Computing, Vienna, Austria). 

## 3. Results

### 3.1. Study Population

Our cohort consisted of 199 patients: the Pentaglobin group included 95 patients, whereas the control group consisted of 104 patients. The baseline patient characteristics of the 199 patients are summarized in Table 1.

The underlying diseases were malignant in 86 children (90%) versus 81 (78%) and nonmalignant in the remaining 9 (10%) versus 23 (22%) in the Pentaglobin group and the control group, respectively. In the Pentaglobin group, 15 children (16%) underwent autologous HSCT, whereas 80 (84%) were given an allogeneic stem cell transplant. The donor’s type was a matched unrelated in 41 cases (51%), a matched related in 20 (25%), and a haploidentical in 19 (24%). In the control group, 15 children underwent autologous HSCT (14%), whereas 89 were given an allogeneic HSCT. A matched unrelated donor was used in 38 cases (43%), a matched related donor in 33 (37%), and a haploidentical donor in 18 (20%). 

Pre-transplant immunological status of patients shows a statistically significant difference in baseline lymphocyte count between the two study groups (*p* < 0.0001). In contrast, we did not find significant differences between the baseline IgG and IgM values comparing both groups. These data are displayed in Figure 2

In the Pentaglobin group, the Pentaglobin^®^ treatment was started when the white blood cell count dropped below 100/μL in all patients. The mean IgM serum concentration was 17.1 mg/dL with ± 11.4 mg/dL of standard deviation (SD) at the start of Pentaglobin^®^ treatment. The mean onset of Pentaglobin^®^ infusion was 6.0 days, with ± 3.9 SD after transplantation, and the mean duration of the treatment was 3.7 days ± 1.2 SD. We compared the differences of sepsis biomarkers, such as C-reactive protein (CRP) and procalcitonin, in both groups, evaluated at mean onset of Pentaglobin^®^ use, and our analysis did not show statistically significant differences. The mean serum CRP concentration (normal range <0.5 mg/dL) was 1.8 mg/dL ± 2.4 SD in the Pentaglobin group versus 2.3 mg/dL ± 2.6 SD in the control group (*p* = 0.1705). Regarding the mean serum procalcitonin levels observed on the same day, they were within the normal range (<0.5 μg/L) in both groups.

### 3.2. Primary Outcome: Six-Month Survival Rate 

After a six-month follow-up period, 23 children (24%) had died in the Pentaglobin group, as compared to 34 patients (33%) in the control group. These data included deaths for primary disease recurrence, which were 47.8% versus 32.4%, and TRM, which were 52.2% versus 67.6%, respectively, in the Pentaglobin Group and the control group (Table 2). A proportion test showed no statistically significant difference between study groups with respect to TRM, save for infections among causes of TRM (*p* = 0.04).

The curve obtained by Kaplan–Meier survival analysis showed no statistically significant difference in the six-month overall survival (OS) rate between the study groups (Figure 3). 

In contrast, Kaplan–Meier survival analysis calculated for deaths due to infectious complications showed a statistically significant difference (*p* = 0.006) (Figure 4). 

Finally, a difference in OS rate after six-month follow-up between the two study groups was evident when excluding mucositis and enteritis among the causes of death, even if statistical significance was not achieved (*p* = 0.061) (Figure 5).

### 3.3. Secondary Outcomes

Box-plot analysis showed a statistically significant difference between the study groups for the number of days with body temperature both for ≥37.3 °C (*p* < 0.001) and ≥38 °C (*p* < 0.001). 

Statistically significant differences between two study groups were found in the number of antibiotics used concurrently during the same infective episode (*p* < 0.0001) and the number of changes of antibiotics because of the failure of treatment (*p* < 0.0001). These box-plot analyses are displayed in Figure 6.

The same analysis performed to compare the number of days of antibiotic therapy and the infective episode recovery rate did not show statistically significant differences. With respect to pathogens identified in blood cultures at sepsis onset, one patient (16.7%) in the Pentaglobin group and three patients (21.4%) in the control group showed *Klebsiella pneumoniae*, the equal percentage of *Pseudomonas aeruginosa* was observed, while two patients (33.3%) in the Pentaglobin group and one patient (7.1%) in the control group showed *Staphylococcus aureus*. Moreover, in the Pentaglobin group, both a case of Brevibacterium and *Serratia marcescens* were detected. Conversely, *Escherichia coli*, *Enterococcus faecalis*, *Enterobacter aerogenes* (two cases for each pathogen), and one case of *Acinetobacter baumannii* was identified in the control group.

Also, after the six-month follow-up period, 33 patients (34.7%) in the Pentaglobin group and 39 (37.1%) in the control group presented with new infectious events following the primary ones detected, but without statistically significant difference between the two groups. Moreover, when the analysis was conducted for a subtype of infection (fungal, opportunistic, viral, or bacterial), a statistically significant difference was observed only for bacterial events. Specifically, only one episode (3%) of bacterial infection was reported in the Pentaglobin group versus eight episodes (20.5%) in the control group (*p* < 0.03).

All secondary outcomes are shown in Table 2.

## 4. Discussion

The twelve-month mortality rate after HSCT remains excessive, especially TRM, which is about 10%.

Notably, with respect to the mortality rate related to disease progression, many new therapeutic strategies have been developed in recent years, such as cellular therapy with cytokine-induced killer cells (CIK) and chimeric antigen receptor T (Car-T) cell therapy or the use of monoclonal antibody therapy such as inotuzumab, blinatumomab, and brentuximab [15]. However, the introduction of new diagnostic tools and amelioration of the current treatment protocols for infectious complications have so far resulted in little comparable improvement. Hence, it has been difficult to obtain a drop in TRM, especially for mortality related to infections developed through the aplasia phase.

Among the new diagnostic tools and improved treatment protocols is the panfungal real-time PCR assay, allowing amplification of any fungal DNA, significantly reducing the time for diagnosis, compared to standard procedures, such as blood culture and histopathological examination [16]. Another recent lab approach is FilmArray, a fast, accurate, molecular diagnostic testing to detect bacteria, viruses, fungi or parasites in biological samples, while also providing information concerning antibiotic-resistant genes [17].

In addition, therapeutic strategies have been developed as a result of the identification of new antimicrobial molecules, overcoming drug-resistance problems, and through the continuous infusion of time-dependent antibiotics, which improves drug safety and efficacy by reducing administered dosage and the emergence of resistant clones [18,19]. Development and use of therapeutic drug monitoring optimize pharmacological therapy for the individual patient, assisting in planning dosage variation, and managing a possible therapeutic failure [20].

In spite of these improvements, infections remain a common complication in patients undergoing HSCT, especially in the aplasia phase. Accordingly, the main effort of clinicians is to prevent the clinical evolution of infection to sepsis and so to reduce infectious disease-related mortality. Pentaglobin^®^, an IgM-enriched immunoglobulin preparation, whose IgM component is responsible for endotoxin antibody activity, seems to be the best therapeutic option to control infectious disease-related mortality in transplant recipients [21]. Several studies have shown a statistically significant mortality rate decrease in cohorts of septic newborns, children, and adolescents undergoing Pentaglobin^®^ therapy. The advantage of Pentaglobin^®^ administration has been most evident for infections caused by Gram-negative pathogens [22,23]. To date, however, there are no studies in the available literature about early adjuvant use of Pentaglobin^®^ for the treatment of patients undergoing HSCT.

The only study previously reported for the pediatric cohort, which compared intravenous immunoglobulin prophylactic use to Pentaglobin^®^, was administered to patients within 100 days after allogeneic HSCT. This study showed no differences between two groups concerning neutropenia, number of hospitalization days, total days of fever, number of consecutive detected infectious events, acute GVHD onset, VOD onset, and side effects [24].

On the other hand, in the adult population, Poynton et al. reported prophylactic use of Pentaglobin^®^ compared to placebo in 63 patients undergoing either autologous or allogeneic HSCT. Their study showed a decreased risk of infection-related mortality in patients who received Pentaglobin^®^ therapy. Also, endotoxemia was lowered in a statistically significant way in the Pentaglobin^®^ group [25].

In another study, Jackson et al. tested blood endotoxemia levels in 1000 plasma samples obtained from adult infected transplant patients. The authors concluded that almost 70% of fevers of uncertain origin were associated with high blood endotoxemia and that the endotoxemia level was lower in patients who underwent Pentaglobin^®^ therapy as compared to the Control group [26].

Our analysis demonstrates that early adjuvant use of Pentaglobin^®^ through aplasia period leads to statistically significant decrease of days of temperature increase, defined either as T°C ≥ 37.3 or T°C ≥ 38, in addition to a reduced infection-related mortality rate compared to the Control group. These data confirmed previous results obtained by Behre et al., who observed that early use of Pentaglobin^®^ in neutropenic patients with suspected Gram-negative sepsis-induced blood anti-endotoxin antibodies and decreased the mortality rate [27].

Transplant recipients through aplasia phase present with a lowered level of anti-endotoxin antibody as compared to donors, especially IgM. Therefore, the use of Pentaglobin^®^ in these patients causes an increased antibody level able to combat high endotoxin levels induced by mucositis, chemotherapy, radiation therapy, and acute GVHD [26].

In addition, our results showed that the Pentaglobin group required fewer antibiotics to control infectious complications and fewer antibiotic therapy changes following first-line drug failure as compared to the control group. Our data support the view that Pentaglobin^®^ early adjuvant use, in addition to standard protocols of antibiotic treatment, might significantly contribute to infection control.

Our study did not show statistically significant differences in total days on antibiotic treatment, perhaps because, in aplasia phase, the patients usually undergo antibiotic therapy until complete hematological reconstitution is accomplished.

Lastly, our study evaluated a number of new bacterial, viral, fungal, and opportunistic infections following the primary one detected. The results did not show a statistically significant difference between the two groups, save for bacterial ones, reduced in Pentaglobin group, in accordance with previously published data [26,27].

Some limitations of this study should be considered. It is a retrospective, monocentric study, with a relatively short follow-up period for outcomes. However, the relatively large selected sample allowed us to get satisfactory results concerning Pentaglobin^®^ use for the treatment of bacterial infections in patients undergoing HSCT. Due to these considerations, further investigations, especially randomized controlled trials, could be valuable in defining real drug efficacy in these patients.

## 5. Conclusions

Pentaglobin^®^ early adjuvant use in the treatment of the patients undergoing HSCT, who show temperature rise during aplasia phase, seems to produce a significant decrease in the rate of infection-associated TRM. Therefore, a prospective multicenter randomized controlled trial should be conducted to define real Pentaglobin^®^ efficacy in patients undergoing HSCT.

## Figures and Tables

**Figure 1 jcm-09-01592-f001:**
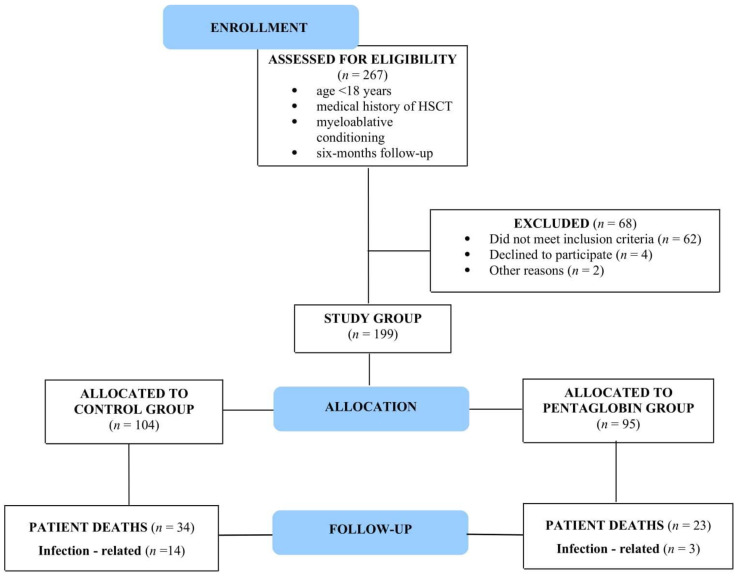
The flowchart shows the number of hematopoietic stem cell transplantation (HSCT) recipients who were screened, enrolled, and included in the final analyses.

**Figure 2 jcm-09-01592-f002:**
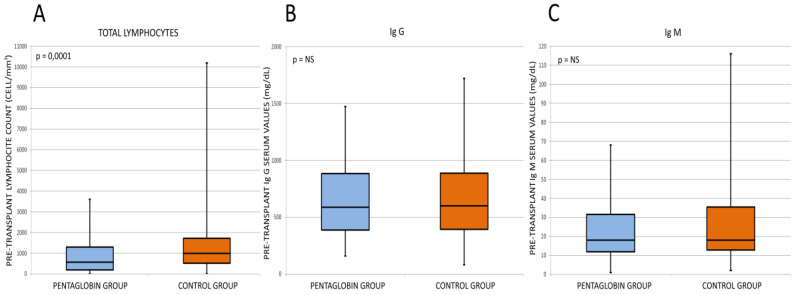
Pre-transplant immunological status of patients in two study groups.

**Figure 3 jcm-09-01592-f003:**
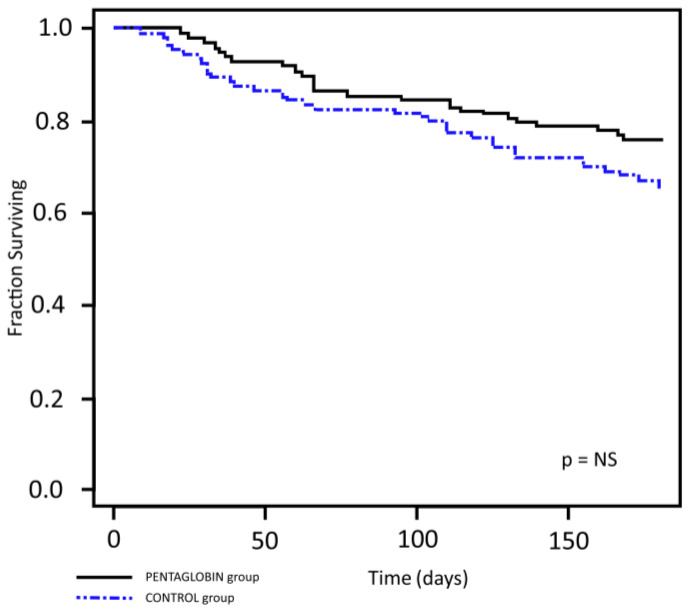
Kaplan-Meier curves for overall survival (OS) in the Pentaglobin group and the control group.

**Figure 4 jcm-09-01592-f004:**
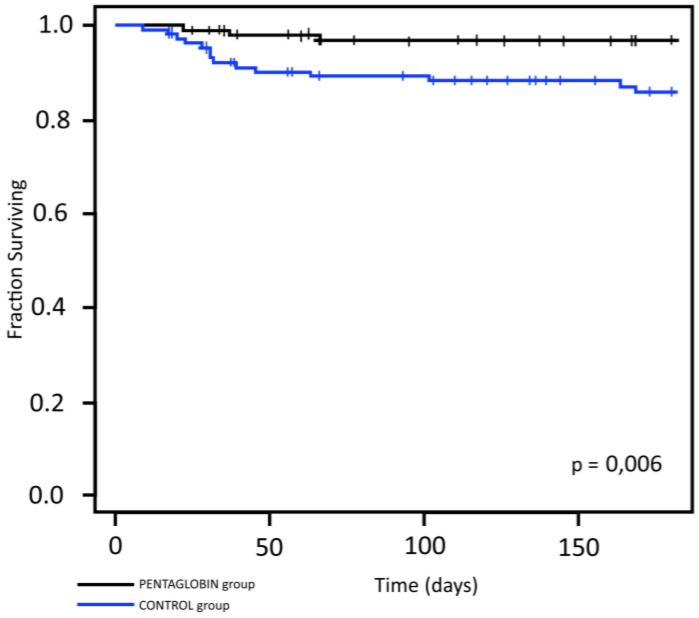
Kaplan–Meier survival curves in two groups calculated for deaths due to infection.

**Figure 5 jcm-09-01592-f005:**
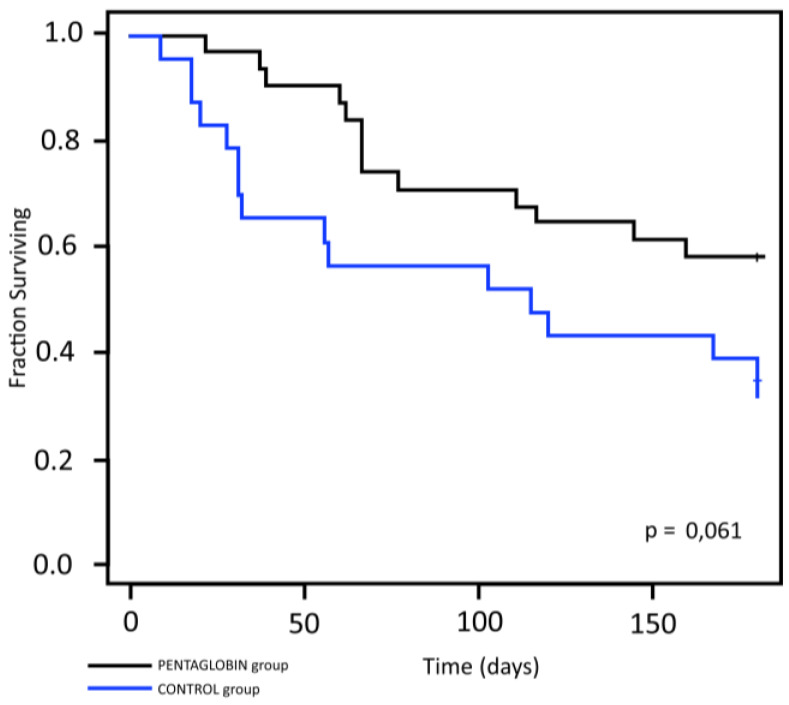
Kaplan–Meier survival curves in two groups calculated for deaths, excluding mucositis and enteritis.

**Figure 6 jcm-09-01592-f006:**
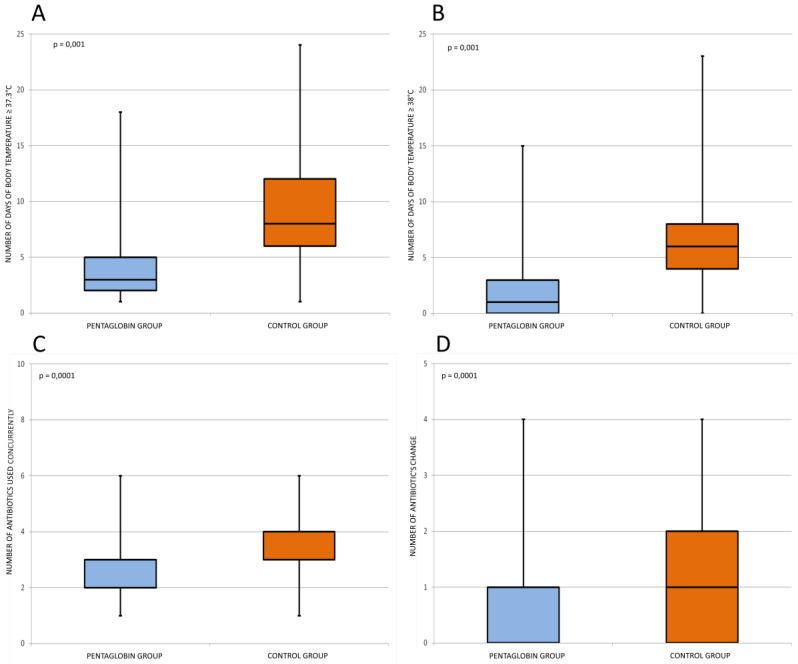
Box plots of secondary outcomes. Box-plot analysis showed a statistically significant difference (*p* < 0.001) between the study groups for the number of days with body temperature ≥37.3 °C (**A**) and ≥38 °C (**B**). Statistically significant differences (*p* < 0.0001) between two study groups were found in the number of antibiotics used (**C**) and the number of changes of antibiotics because of the failure of treatment (**D**).

**Table 1 jcm-09-01592-t001:** Patient demographics, clinical, and HSCT features.

Variables	Pentaglobin Group	Control Group
Number of patients (%)	95 (48)	104 (52)
Sex, number (%)		
Male	64 (67.4)	69 (66.3)
Female	31 (32.6)	35 (33.7)
Age at transplant, years, mean (± SD)	9.6 (±5.1)	8.0 (±5.6)
Underlying disease, number (%):		
Acute lymphoblastic leukaemia	42 (44)	32 (31)
Acute myeloid leukaemia	16 (17)	20 (19)
Myelodysplastic syndrome	8 (8)	10 (10)
Solid tumour	20 (21)	19 (18)
Other	9 (10)	23 (22)
Disease stage, number (%) *		
Early	22 (33)	25 (40)
Intermediate	25 (38)	20 (32)
Late	19 (29)	17 (28)
Type of transplant, number (%)		
Autologous	15 (16)	15 (14)
Allogeneic:	80 (84)	89 (86)
*Matched related donor*	20 (25)	33 (37)
*Matched unrelated donor*	41 (51)	38 (43)
*Haploidentical donor*	19 (24)	18 (20)
Myeloablative conditioning, number (%)		
MCHT-based	11 (12)	24 (23)
TBI-based	84 (88)	80 (77)
Graft source, number (%)		
Bone marrow	70 (74)	84 (81)
Peripheral blood stem cells	25 (26)	20 (19)

MCHT, myeloablative chemotherapy; TBI, total body irradiation; SD, standard deviation. * Disease stage was defined according to previously published classification. This classification is applied to patients with acute leukemia and myelodysplastic syndrome only ^14^.

**Table 2 jcm-09-01592-t002:** Transplant-related outcomes. *p*-values refer to two-tailed Fisher exact tests if relative to 2 × n contingency tables, or to Mann–Whitney rank sum test if comparing values of continuous variables between two groups.

Variables	Pentaglobin Group	Control Group	*p*-Value
	(95 Patients)	(104 Patients)	
**Type of infection, number (%):**			
Sepsis	6 (6.3)	14 (13.5)	0.104
Pneumonia	7 (6.7)	13 (13.7)	0.106
Pulmonary aspergillosis	1 (1.0)	1 (0.95)	1
Brain abscess	4 (4.2)	0	0.049
Soft tissue infections	1 (0.95)	5 (5.3)	0.104
Cholangitis and biliary tract infection	0	3 (3.2)	0.105
Mucositis- enterocolitis	62 (65.3)	81 (77.9)	0.084
Other	1 (1.0)	0	0.475
**Days with body temperature ≥ 37.3 °C, mean (± SD)**	4.2 (3.1)	8.9 (4.9)	<0.0001
**Days with body temperature ≥ 38.0 °C, mean (± SD)**	2.1 (2.8)	6.6 (4.3)	<0.0001
**Days on antibiotic treatment, mean (± SD)**	15.1 (5.4)	16.8 (7.2)	0.165
**Number of antibiotics used, mean (± SD)**	2.7 (1.0)	3.3 (1.0)	<0.0001
**Antibiotic changes during infective episode, mean (± SD)**	0.6 (0.8)	1.1 (1.0)	<0.0001
**Infective episode recovery, number (%)**	93 (97.9)	96 (92.3)	0.105
**Consecutive infections, number (%):**	33 (34.7)	39 (37.1)	0.769
Bacterial	1 (3.0)	8 (20.5)	0.037
Fungal	4 (12.1)	5 (12.8)	1
Opportunistic	1 (3.0)	3 (7.7)	0.623
Virus	27 (81.8)	23 (59.0)	0.328
**Overall survival at 6 months, number (%)**	72 (76)	70 (67)	0.081
**Cause of death, number (%):**			
Disease progression	11 (47.8)	9 (32.4)	0.431
Transplant-related mortality	12 (52.2)	23 (67.6)	0.086
*of which infection*	3 (25)	14 (60.8)	0.04

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
