# Peer review of "Pentaglobin® Efficacy in Reducing the Incidence of Sepsis and Transplant-Related Mortality in Pediatric Patients Undergoing Hematopoietic Stem Cell Transplantation: A Retrospective Study"

_jcm, 2020, doi:10.3390/jcm9051592_

Round 1

Reviewer 1 Report

The issue of the impact of Pentaglobin on the mortality of infection-related mortality associated with hematopoietic stem cell transplantation is very relevant for clinical practice. However, the important pieces of information are missing, listed below. They should be provided in the revision process.

  • Why patients who received started administration more than 12 hours after their first temperature rise were excluded from the study? I would suggest the additional analysis of this patients category to determine a strength/limitation Pentaglobin effect.
  • Due to the retrospective character of the study, it should be given supplemental data on CRP, procalcitonin concentrations to prove comparability between Penta globin and control groups.
  • It should be specified ay what time post-procedure Pentaglobin was applied. IgM has half-life 5-6 days, whereas curves on fig. 4 diverge around one month.
  • Viral infections were more frequent in the Penataglobin group. It could be the cause of lesser antibiotics number usage in this group compared with control.
  • Finally, information on the identified pathogens should be added.

Author Response

Revision: R1 of the manuscript jcm-792797 entitled “Pentaglobin ® efficacy in reducing the incidence of sepsis and transplant-related mortality in pediatric patients undergoing hematopoietic stem cell transplantation: a retrospective study ”.

The manuscript was revised and below we have addressed point to point the issues raised in the comments.

Reply to Reviewer #1:

We want to thank  the Reviewer for his thoughtful and insightful comments.

1)Why patients who received started administration more than 12 hours after their first temperature rise were excluded from the study? I would suggest the additional analysis of this patients category to determine a strength/limitation Pentaglobin effect.

- We appreciate this referee’s question. The aim of the current study is to test the use of Pentaglobin® as early adjuvant treatment for febrile and subfebrile pediatric patients undergoing HSCT, to determine if this therapeutic approach might improve clinical outcome, decreasing the early TRM rate; so, with respect to our therapeutic protocol for febrile and subfebrile pediatric patients through aplasia phase, antibiotic treatment  is started  within 12 hours from first body temperature rise ≥ 37.3 Celsius degrees (°C, 99.1°F), in addition to Pentaglobin ® in the case group. So, we don’t have available data related to the patient cohort which started antibiotic therapy 12 hours after first temperature rise.

2)Due to the retrospective character of the study, it should be given supplementale data on CRP, procalcitonin concentration to prove comparability between Pentaglobin and Control group.

- We fully agree with the Reviewer’s comment and we mentioned our reply in the section 3.1., lines 201-207.

3) It should be specified ay what time post-procedure Pentaglobin was applied. IgM has half-life 5-6 days, whereas curves on fig. 4 diverge around one month.

- We fully agree with the Reviewer’s comment. In our study Kaplan Meier shows survival rate after a 6-month follow-up period from transplant event, so time 0 represents day of HSCT while start of Pentaglobin use is different for patients (different times) and related to clinical symptoms and signs suggestive for sepsis onset through aplasia phase, according to the aim of the current study. With respect to these explanations, reported curves in figure 4 diverged later respect to IgM half-life. Pentaglobin ® administration has been reported in Material and Methods of the current study, section 2.3.

4) Viral infections were more frequent in the Penataglobin group. It could be the cause of lesser antibiotics number usage in this group compared with control.

- We appreciate this referee’s comment. With respect to the current study, data related to viral infections detected in the Pentaglobin group versus the Control group can be considered as observational data without statistical significance, as reported in the Table 2, line 261 (p= 0.328).

5) Finally, information on the identified pathogens should be added.

- We fully agree with the Reviewer’s comment and we have mentioned our reply in the section 3.2, lines 244-252.

Reviewer 2 Report

This manuscript describes the effect of administration of Pentaglobin for lowering sepsis through apalasia phase with respect to transplant-related mortality. They show early use of Pentaglobin through apalasia period leads to significant decrease infection related mortality rate compared to control group. In introduction section, it might be necessary to describe what is Pentaglobin and what is the scientific significance of this study. The characteristics of antibiotics or other adjuvants which might be necessary in HSCT. 

Author Response

Reply to Reviewer #2:

We want to thank the Reviewer for his thoughtful and insightful comments.

“..It might be necessary to describe what is Pentaglobin and what is the scientific significance of this study”.

- We fully agree with the Reviewer’s comment and we have mentioned our reply in the section 1., lines 88-91 and lines 94-96.

“..The characteristics of antibiotics or other adjuvants which might be necessary in HSCT.”

-We appreciate this referee’s comment. Our Transplant Unit protocol for antibiotic treatment in transplant recipients through aplasia phase is standardized with respect to the national guidelines of GITMO. We have mentioned our reply in the section 2.3. of the current study (specifically, lines 153-156).

Again we want to thanks the reviewers for their comments. Notice that all the corrections are in bold type (mentioned lines in this letter).

We hope the manuscript is now improved. However, we are fully available for other changes if requested.

Enclosed is a copy of the revised manuscript with highlights.

Round 2

Reviewer 1 Report

The manuscript is significantly improved. My remarks are appropriately considered.